# VCT: A Video Compression Transformer

**Fabian Mentzer**
Google Research
mentzer@google.com

**George Toderici**
Google Research
gtoderici@google.com

**David Minnen**
Google Research
dminnen@google.com

**Sung Jin Hwang**
Google Research
sjhwang@google.com

**Sergi Caelles**
Google Research
scaelles@google.com

**Mario Lucic**
Google Research
lucic@google.com

**Eirikur Agustsson**
Google Research
eirikur@google.com

## Abstract

We show how transformers can be used to vastly simplify neural video compression. Previous methods have been relying on an increasing number of architectural biases and priors, including motion prediction and warping operations, resulting in complex models. Instead, we independently map input frames to representations and use a transformer to model their dependencies, letting it predict the distribution of future representations given the past. The resulting video compression transformer outperforms previous methods on standard video compression data sets. Experiments on synthetic data show that our model learns to handle complex motion patterns such as panning, blurring and fading purely from data. Our approach is easy to implement, and we release code to facilitate future research.

## 1 Introduction

Neural network based video compression techniques have recently emerged to rival their non-neural counter parts in rate-distortion performance [*e.g.*, 1, 17, 30, 42]. These novel methods tend to incorporate various architectural biases and priors inspired by the classic, non-neural approaches. While many authors tend to draw a line between "hand-crafted" classical codecs and neural approaches, the neural approaches themselves are increasingly "hand-crafted", with authors introducing complex connections between the many sub-components. The resulting methods are complicated, challenging to implement, and constrain themselves to work well only on data that matches the architectural biases. In particular, many methods rely on some form of motion prediction followed by a warping operation [*e.g.*, 1, 17, 19, 23, 42, 41]. These methods warp previous reconstructions with the predicted flow, and calculate a residual.

In this paper, we replace flow prediction, warping, and residual compensation, with an elegantly simple but powerful transformer-based temporal entropy model. The resulting video compression transformer (VCT) outperforms previous methods on standard video compression data sets, while being free from their architectural biases and priors. Furthermore, we create synthetic data to explore the effect of architectural biases, and show that we compare favourably to previous approaches on the types videos that the architectural components were designed for (panning on static frames, or blurring), despite our transformer not relying on any of these components. More crucially, we outperform previous approaches on videos that have no obvious matching architectural component (sharpening, fading between scenes), showing the benefit of removing hand-crafted elements and letting a transformer learn everything from data.

We use transformers to compress videos in two steps (see Fig. 1): First, using *lossy* transform coding [3], we map frames $x_i$ from image space to quantized representations $y_i$, *independently for each frame*. From $y_i$ we can recover a reconstruction $\hat{x}_i$. Second, we let a transformer leverage

36th Conference on Neural Information Processing Systems (NeurIPS 2022).

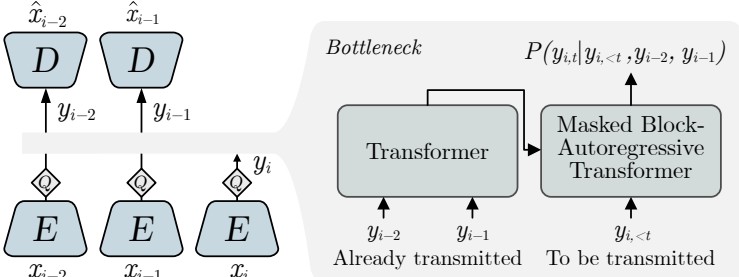

Figure 1: We independently and *lossily* map input frames $x$ into quantized representations $y$. From $y$ we can recover a reconstruction $\hat{x}$. To store $y_i$ with few bits, we use transformers to model temporal dependencies and to predict a distribution for $y_i$ given previously transmitted representations. We use $P$ to *losslessly* compress the quantized $y_i$ using entropy coding. The better the transformer predicts $P$, the fewer bits are required to store $y_i$. We note that we have no hard-coded components such as motion prediction or warping.

temporal redundancies to model the distributions of the representations. We use these predicted distributions to *losslessly* compress the quantized $y_i$ using entropy coding [43, Sec 2.2.1]. The better the transformer predicts the distributions, the fewer bits are required to store the representations.

This setup avoids complex state transitions or warping operations by letting the transformer learn to leverage arbitrary relationships between frames. As a bonus, we get rid of temporal error propagation by construction since the reconstruction $\hat{x}_i$ does not depend on previous reconstructions. Contrast with warping-based approaches, where $\hat{x}_i$ is a function of the warped $\hat{x}_{i-1}$ meaning that any visual errors in $\hat{x}_i$ will be propagated forward and require additional bits to correct with residuals.

VCT is based on the original language translation transformer [35]: We can view our problem as "translating" two previous representations $y_{i-2}, y_{i-1}$ to $y_i$. However, there are various challenges in the way of directly applying the NLP formulation. Consider a 1080p video frame; using a typical neural image compression encoder [4] that downscales by a factor 16 and has 192 output channels, a (1080, 1920, 3)-dimensional input frame is mapped to a (68, 120, 192)-dimensional feature representation leading to approximately 1.6 million symbols. Naively correlating all of these symbols to all symbols in a previous representation would yield a 1.6M×1.6M-dimensional attention matrix. To address this computationally impractical problem, we introduce independence assumptions to shrink the attention matrix and enable parallel execution on subsets of the symbols.

Our model is easy to implement with contemporary machine learning frameworks, and we provide an extensive code and model release to allow future work to build on this direction.[1]

## 2   Related Work

Transformers were initially proposed for machine translation [35], where an encoder-decoder structure was used to obtain state-of-the-art results. This led to a wide range of follow-up research, and state-of-the-art natural language processing (NLP) models are still based on transformers [*e.g.*, 6, 8, 7, 11]. Motivated by these advancements, Dosovitski *et al.* [10] replaced CNNs with a transformer-based architecture to achieve state-of-the-art results in image classification, which in turn spurred more exploration of transformers in the computer vision community including both image tasks [*e.g.*, 22, 37, 45] as well as video analysis [*e.g.*, 2, 5, 12, 28, 32].

Recently, transformers were incorporated into neural *image* compression models. Qian *et al.* [29] replaced the autoregressive hyperprior [26] with a self-attention stack, and Zhu *et al.* [46] replaced all convolutions in the standard approach [4, 27] with Swin Transformer [22] blocks.

Neural *video* compression remains CNN-based. After initial work used frame interpolation [39, 9], Lu *et al.* [23] followed the more traditional approach of predicting optical flow between the previous reconstruction and the input, transmitting a compressed representation of the flow, and also transmitting a residual image to correct visual errors after warping. Many papers extended this

---

[1]https://goo.gle/vct-paper

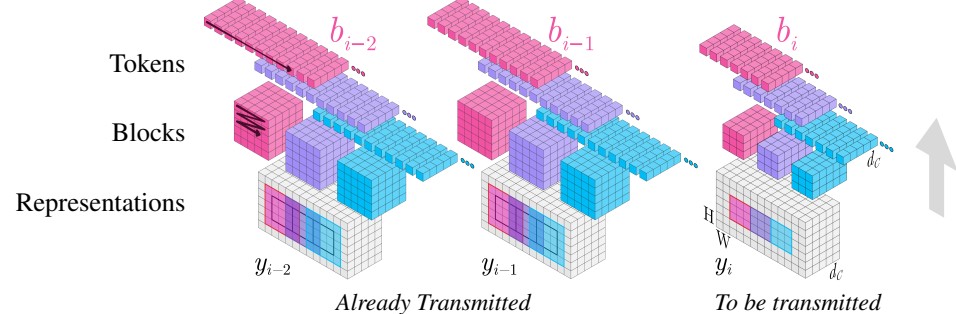

Figure 2: From representations to tokens. We essentially use a sliding window to split the current representation $y_i$ into *non-overlapping* $w_c \times w_c$ blocks, and previous representations $y_{i-2}, y_{i-1}$ into *overlapping* $w_p \times w_p$ blocks with stride $w_c$ ($w_p > w_c$). We flatten blocks spatially (raster-scan order, see left arrows) to obtain tokens for the transformer, which remain $d_C$-dimensional since they are just a different view of $y_i$. We show $w_c{=}3, w_p{=}5, d_C{=}5$, but we use $w_c{=}4, w_p{=}8, d_C{=}192$ in practice.

approach, for example Agustsson *et al.* [1] introduced the notion of a flow predictor that also supports blurring called "Scale Space Flow" (SSF), which became a building block for other approaches [42, 30]. Rippel *et al.* [30] achieved state-of-the-art results by using SSF and more context to predict flow. RNNs and ConvLSTMs were used to build recurrent decoders [13] or entropy models [41].

Some work does not rely on pixel-space flow: Habibian *et al.* [14] used a 3D autoregressive entropy model, FVC [17] predicted flow in a $2\times$ downscaled feature space, and Liu *et al.* [20] used a ConvLSTM to predict representations which are transmitted using an iterative quantization scheme. DCVC [19] estimated motion in pixel space but performed residual compensation in a feature space. Liu *et al.* [21] also losslessly encoded frame-level representations, but rely on CNNs for temporal modelling. Finally, recent work employed GAN losses to increase realism [24, 40].

## 3 Method

### 3.1 Overview and Background

**Frame encoding and decoding**   A high-level overview of our approach is shown in Fig. 1. We split video coding into two parts. First, we independently encode each frame $x_i$ into a *quantized* representation $y_i{=}\lfloor E(x_i) \rceil$ using a CNN-based image encoder $E$ followed by quantization to an integer grid. The encoder downscales spatially and increases the channel dimension, resulting in $y_i$ being a $(H, W, d_C)$-dimensional feature map, where $H, W$ are $16\times$ smaller than the input image resolution. From $y_i$, we can recover a reconstruction $\hat{x}_i$ using the decoder $D$. We train $E, D$ using standard neural image compression techniques to be lossy transforms reaching nearly any desired distortion $d(x_i, \hat{x}_i)$ by varying how large the range of each element in $y_i$ is. For now, let us assume we have a pair $E, D$ reaching a fixed distortion.

**Naive approach**   After having *lossily* converted the sequence of input frames $x_i$ to a sequence of representations $y_i{=}\lfloor E(x_i) \rceil$, one can naively store all $y_i$ to disk *losslessly*. To see why this is sub-optimal, let each element $y_{i,j}$ of $y_i$ be a symbol in $\mathcal{S} = \{-L, \ldots, L\}$. Assuming that all $|\mathcal{S}|$ symbols appear with equal probability, *i.e.*, $P(y_{i,j}) = 1/|\mathcal{S}|$, one can transmit $y_i$ using $H \cdot W \cdot d_C \cdot \log_2 |\mathcal{S}|$ bits. Using a realistic $L{=}32$, this implies that we would need $9\,\mathrm{MB}$, or $\approx 2\mathrm{Gbps}$ at 30fps, to encode a single HD frame (where $H \cdot W \cdot d_C \approx 1.6\mathrm{M}$, see Introduction). While arguably inefficient, this is a valid compression scheme which will result in the desired distortion. The aim of this work is to improve this scheme by approximately two orders of magnitude.

**An efficient coding scheme**   Given a probability mass function (PMF) $P$ estimating the true distribution $Q$ of symbols in $y_i$, we can use entropy coding (EC) to transmit $y_i$ with $H \cdot W \cdot d_C \cdot \mathbb{E}_{y \sim Q(y_i)}[-\log_2 P(y)]$ bits.[2]   By using EC, we can encode more frequently occurring values with fewer bits, and hence improve the efficiency. Note that the expectation term representing the average bit count corresponds to the cross-entropy of $Q$ with respect to $P$. Our main idea is to parameterize $P$

---

[2]Consistent with neural compression literature but in contrast to Information Theory, we use $P$ for the model.

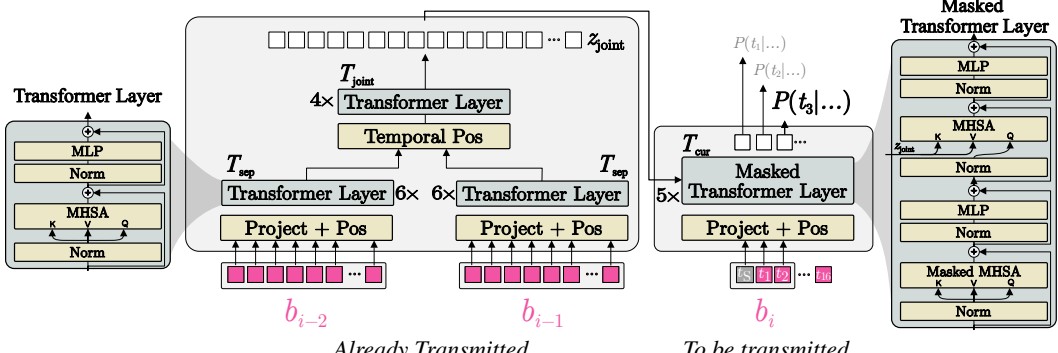

Figure 3: The transformer operates on the pink set of blocks/tokens $b_{i-2}, b_{i-1}, b_i$ (obtained as shown in Fig. 2). We first extract temporal information $z_{\text{joint}}$ from already transmitted blocks. $T_{\text{cur}}$ is shown predicting $P(t_3|t_S, t_1, t_2, z_{\text{joint}})$, where $t_S$ is a learned start token.

as a conditional distribution using very flexible transformer models, and to minimize the cross-entropy and thus maximize coding efficiency. We emphasize that we use $P$ for lossless EC, we do not sample from the model to transmit data. Even if the resulting model of $P$ is sub-optimal, $y_i$ can still be stored losslessly, albeit inefficiently.

Why would one hope to do better than the uniform distribution over $y_i$? In principle, the model should be able to exploit the temporal redundancy across frames, and the spatial consistency within frames.

## 3.2 Transformer-based Temporal Entropy Model

To transmit a video of $F$ frames, $x_1, \ldots, x_F$, we first map $E$ over each frame obtaining quantized representations $y_1, \ldots, y_F$. Let's assume we already transmitted $y_1, \ldots, y_{i-1}$. To transmit $y_i$, we use the transformer to predict $P(y_i|y_{i-2}, y_{i-1})$. Using this distribution, we entropy code $y_i$ to create a compressed, binary representation that can be transmitted. To compress the full video, we simply apply this procedure iteratively, letting the transformer predict $P(y_j|y_{j-2}, y_{j-1})$ for $j \in \{1, \ldots, F\}$, padding with zeros when predicting distributions for $y_1, y_2$. The receiver follows the same procedure to recover all $y_j$, *i.e.*, it iteratively calculates $P(y_j|y_{j-2}, y_{j-1})$ to entropy decode each $y_j$. After obtaining each representation, $y_1, y_2, \ldots, y_F$, the receiver generates reconstructions.

**Tokens** When processing the current representation $y_i$, we split it spatially into *non-overlapping* blocks with size $w_c \times w_c$ as shown in Fig. 2. Previous representations $y_{i-2}, y_{i-1}$ become corresponding *overlapping* $w_p \times w_p$ blocks (where $w_p > w_c$) to provide both temporal and spatial context for predicting $P(y_i|y_{i-2}, y_{i-1})$. Intuitively, the larger spatial extent provides useful context to predict the distribution of the current block. Note that all blocks span a relatively large spatial region in image space due to the downscaling convolutional encoder $E$. We flatten each block spatially (see Fig. 2) to obtain tokens for the transformers. The transformers then run independently on corresponding blocks/tokens, *i.e.*, tokens of the same color in Fig. 2 get processed together, trading reduced spatial context for parallel execution.[3]

This independence assumption allows us to focus on a single set of blocks, *e.g.*, the pink blocks in Fig. 2. In the following text and in Fig. 3, we thus show how we predict distributions for the $w_c^2 = 16$ tokens $t_1, t_2, \ldots, t_{16}$ in block $b_i$, given the $2w_p^2 = 128$ tokens from the previous blocks $b_{i-2}, b_{i-1}$.

**Step 1: Temporal Mixer** We use two transformers to extract temporal information from $b_{i-2}, b_{i-1}$. A first transformer $T_{\text{sep}}$ operates separately on each previous block. Then, we concatenate the outputs in the token dimension and run the second transformer, $T_{\text{joint}}$, on the result to mix information across time. The output $z_{\text{joint}}$ is $2w_p^2$ features, containing everything the model "knows" about the past.

**Step 2: Within-Block-Autoregression** The second part of our method is the masked transformer $T_{\text{cur}}$, which predicts PMFs for each token using auto-regression within the block. We obtain a

---

[3]As a side benefit, the number of tokens for the transformers is not a function of image resolution, unlike ViT-based approaches [10].

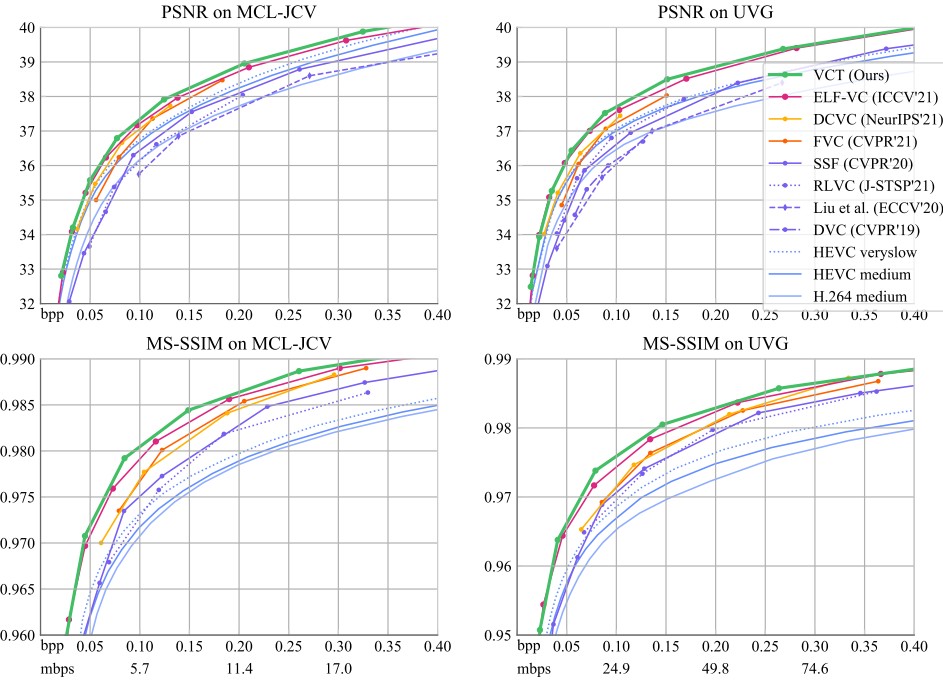

Figure 4: Comparing rate-distortion on MCL-JCV ($\approx$27FPS) and UVG (120FPS). We report bits per pixel (bpp) and megabits per second (mbps). For MS-SSIM, we only show methods optimized for it (using *–tune ssim* for HEVC/H264). *App. A.5 shows a large version of these plots.*

powerful model by conditioning $T_{\text{cur}}$ on $z_{\text{joint}}$ as well as already transmitted tokens within the block. For entropy coding, both the sender and the receiver must be able to obtain exactly the same PMFs, *i.e.*, $T_{\text{cur}}$ must be causal and start from a known initialization point. For the latter, we learn a *start token $t_S$*.

To send the tokens, we first obtain $z_{\text{joint}}$. After that, we feed $[t_S]$ to $T_{\text{cur}}$, obtain $P(t_1|t_S; z_{\text{joint}})$, and use entropy coding to store the $d_C$ symbols in token $t_1$ into a bitstream using $P(t_1|t_S; z_{\text{joint}})$. Then, we feed $[t_S, t_1]$, obtain $P(t_2|t_1, t_S; z_{\text{joint}})$, store $t_2$ in the bitstream, and so on. The receiver gets the resulting bitstream and can obtain the same distributions, and thereby the tokens, by first feeding $[t_S]$ to $T_{\text{cur}}$, obtaining $P(t_1|t_S; z_{\text{joint}})$, entropy decoding $t_1$ from the bitstream, then feeding $[t_S, t_1]$ to obtain $P(t_2|t_1, t_S; z_{\text{joint}})$, and so on. Fig. 3 visualizes this for $P(t_3|\dots)$.

We run this procedure in parallel over all blocks, and thereby send/receive $y_i$ by running $T_{\text{cur}}$ $w_{\text{c}}^2{=}16$ times. Each run produces $\lceil H/w_{\text{c}} \rceil \cdot \lceil W/w_{\text{c}} \rceil \cdot d_C$ distributions. To ensure causality of $T_{\text{cur}}$ during training, we mask the self-attention blocks similar to [35].

**Independence**  Apart from assuming blocks in $y_i$ are independent, we emphasize that each token is a vector and that we assume the symbols within each token are conditionally independent given previous tokens, *i.e.*, $T_{\text{cur}}$ predicts the $d_C$ distributions required for a token *at once*. One could instead predict a joint distribution over all possible $|\mathcal{S}|^{d_C}$ realisations, use channel-autoregression [27], or use vector quantization on tokens. The latter two are interesting directions for future work. Finally, we note that we do not rely on additional side information, in contrast to, *e.g.*, autoregressive image compression entropy models [26, 27].

### 3.3 Architectures

**Transformers**  As visualized in Fig. 3, all of our transformers are based on standard architectures [35, 10]. We start by projecting the $d_C$-dimensional tokens to a $d_T$-dimensional space ($d_T{=}768$ in our model) using a single fully connected layer, and adding a learned positional embedding. While both $T_{\text{sep}}$ and $T_{\text{joint}}$ are stacks of multi-head self-attention (MHSA) layers, $T_{\text{cur}}$ uses masked "conditional" transformer layers, similar to Vaswani *et al.* [35]: These alternate between masked MHSA layers and MHSA layers that use $z_{\text{joint}}$ as keys (K) and values (V), as shown in Fig. 3. We use 6 transformer layers for $T_{\text{sep}}$, 4 for $T_{\text{joint}}$, and 5 masked transformer layers for $T_{\text{cur}}$. We use 16

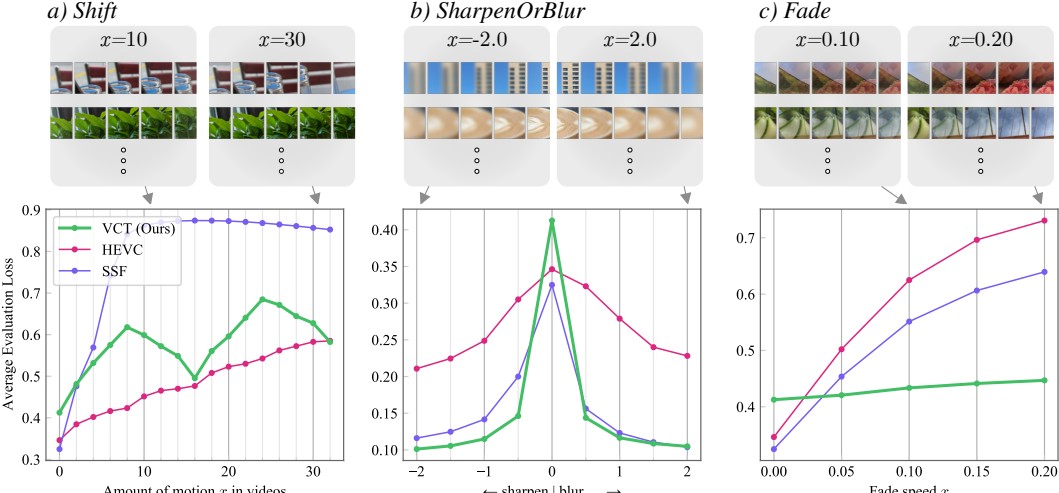

Figure 5: To understand what types of temporal patterns our transformer has learned to exploit, we synthesize videos representing commonly seen patterns. We compare to HEVC, which has built-in support for motion, and SSF, which has built-in support for motion and blurrying. VCT learns to handle all patterns purely from data. We refer to the text for a discussion.

attention heads everywhere. We learn a separate temporal positional embedding to add to the input of $T_{\text{joint}}$.

**Image encoder $E$, decoder $D$**  The image encoder and decoder $E, D$ are not the focus of this paper, so we use architectures based on standard image compression approaches [26, 27]. For the encoder, we use 4 strided convolutional layers, downscaling by a factor $16\times$ in total. For the decoder, we use transposed convolutions and additionally add residual blocks at the low resolutions. We use $d_{ED}=192$ filters for all layers. See App. A.1 for details and an exploration of architecture variants.

### 3.4   Loss and Training Process

The modeling choices introduced in the previous section allow for an efficient training procedure where we decompose the training into three stages, which enables rapid experimentation (Tab. 1). In **Stage I** we train the per-frame encoder $E$ and decoder $D$ by minimizing the rate-distortion trade-off [43, Sec 3.1.1]. Let $\mathcal{U}$ denote a uniform distribution in $[-0.5, 0.5]$. We minimize

$$\mathcal{L}_{\text{I}} = \mathbb{E}_{x \sim p_X, u \sim \mathcal{U}}[\underbrace{-\log p(\tilde{y}+u)}_{\text{bit-rate } r} + \lambda \underbrace{\text{MSE}(x, \hat{x})}_{\text{distortion } d}], \qquad \tilde{y}=E(x), \ \hat{x}=D(\text{round}_{\text{STE}}(\tilde{y})), \quad (1)$$

using $\tilde{y}$ to refer to the unquantized representation, and $x \sim p_X$ are frames drawn from the training set. Intuitively, we want to minimize the reconstruction error under the constraint that we can effectively quantize the encoder output, with $\lambda$ controlling the tradeoff. For *Stage I*, we thus employ the mean-scale hyperprior [26] approach to estimate $p$, the de facto standard in neural image compression, which we discard for later stages.[4] To enable end-to-end training, we also follow [26], adding i.i.d.

---

[4]In short, the hyperprior estimates the PMF of $y$ using a VAE [18], by predicting $p(y|z)$, where $z$ is side information transmitted first. We refer to the paper for details [26].

| | Components trained | Loss | $B$ | $N_F$ | LR | Steps | steps/s |
|---|---|---|---|---|---|---|---|
| Stage I | $E, D$ | $r + \lambda d$ | 16 | 1 | $1\text{E}^{-4}$ | 2M | 100 |
| Stage II | $T_{\text{sep}}, T_{\text{joint}}, T_{\text{cur}}$ | $r$ | 32 | 3 | $1\text{E}^{-4}$ | 1M | 10 |
| Stage III | $T_{\text{sep}}, T_{\text{joint}}, T_{\text{cur}}, E, D$ | $r + \lambda d$ | 32 | 3 | $2.5\text{E}^{-5}$ | 250k | 5 |

Table 1: We split training in three stages for training efficiency (note the steps/s column). $\lambda$ controls the rate-distortion trade-off, $r$ is bitrate, $d$ is distortion, $B$ is batch size, $N_F$ the number of frames.

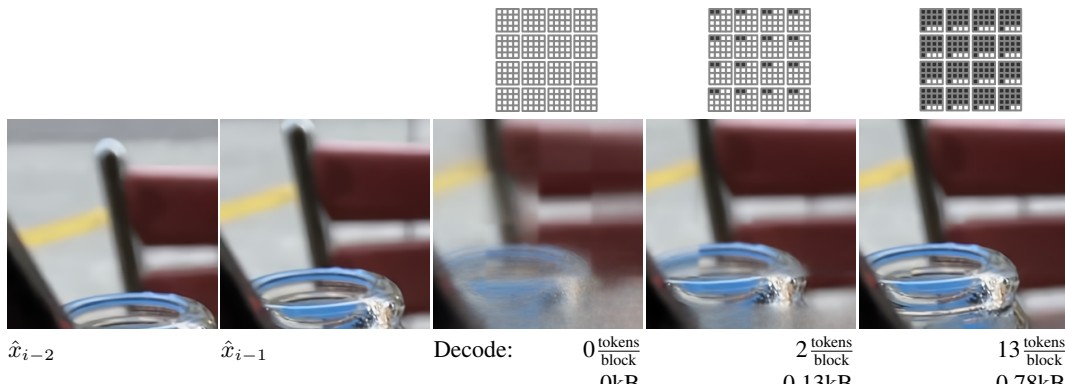

| | | |
|---|---|---|
| $\hat{x}_{i-2}$ | $\hat{x}_{i-1}$ | Decode: |

| $0\frac{\text{tokens}}{\text{block}}$ | $2\frac{\text{tokens}}{\text{block}}$ | $13\frac{\text{tokens}}{\text{block}}$ |
|---|---|---|
| 0kB | 0.13kB | 0.78kB |

Figure 6: Visualizing the sample mean from the block-autoregressive distribution predicted by the transformer, as we decode more and more tokens (see Sec. 5). We show the kilobytes (kB) required to transmit the decoded (gray) tokens. On the left, we see the two previous reconstructions $\hat{x}_{i-2}, \hat{x}_{i-1}$. In the middle, we see what the transformer expects at the current frame, *before decoding any information* (0kB). The next two images shows that as we decode more tokens, the model gets more certain, and the image obtained from the sample mean sharpens. Note that we never sample from the model for actual video coding.

uniform noise $u$ to $\tilde{y}$ when calculating $r$, and using straight-through estimation (STE) [33, 27] for gradients when rounding $\tilde{y}$ to feed it to $D$.

For **Stage II**, we train the transformer to obtain $p$, and only minimize rate:

$$\mathcal{L}_{\text{II}} = \mathbb{E}_{(x_1,x_2,x_3)\sim p_{X_{1:3}}, u\sim\mathcal{U}}[-\log p(\tilde{y}_3 + u|y_1, y_2)] \qquad \tilde{y}_i = E(x), \ y_i = \text{round}(\tilde{y}_i), \qquad (2)$$

where $(x_1, x_2, x_3) \sim p_{X_{1:3}}$ are triplets of adjacent video frames. We assume each of the $d_C$ unquantized elements in each token follow a Gaussian distribution, $p \sim \mathcal{N}$, and let the transformer predict $d_C$ means and $d_C$ scales per token. Finally, we finetune everything jointly in **Stage III**, adding the distortion loss $d$ from Eq. 1 to Eq. 2.

We note that it is also possibe to train the model **from scratch** and obtain even better performance, see App. A.2.

To obtain a discrete PMF $P$ for the quantized symbols (for entropy coding), we again follow standard practice [4], convolving $p$ with a unit-width box and evaluating it at discrete points, $P(y) = \int_{u\in\mathcal{U}} p(y+u)du, y \in \mathbb{Z}$ [see, *e.g.*, 43, Sec. 3.3.3, for details]. To train, we use random spatio-temporal crops of $(B, N_F, 256, 256, 3)$ pixels, where $B$ is the batch size, and $N_F$ the number of frames (values are given in Tab. 1). We use the linearly decaying learning rate (LR) schedule with warmup, where we warmup for 10k steps and then linearly decay from the LR shown in the table to $1\text{E}^{-5}$. *Stage I* is trained using $\lambda=0.01$. To navigate the rate-distortion trade-off and obtain results for multiple rates, we fine-tune 9 models in *Stage III*, using $\lambda=0.01 \cdot 2^i, i\in\{-3, \ldots, 5\}$. We train all models on 4 Google Cloud TPUv4 chips.

### 3.5 Latent Residual Predictor (LRP)

To further leverage the powerful representation that the transformer learns, we adapt the "latent residual predictor" (LRP) from recent work in image compression [27]: The final features $z_{\text{cur}}$ from $T_{\text{cur}}$ have the same spatial dimensions as $y_i$, and contain everything the transformer knows about the current and previous representations. Since we have to compute them to compute $P$, they constitute "free" extra features that are helpful to reconstruct $\hat{x}_i$. We thus use $z_{\text{cur}}$ by feeding $y_i' = y_i + f_{\text{LRP}}(z_{\text{cur}})$ to $D$ (we enable this in *Stage III*), where $f_{\text{LRP}}$ consists of a $1\times1$ convolution mapping from $d_T$ to $d_{ED}$ followed by a residual block. We note that this implies that $\hat{x}_i = D(y_i')$ indirectly depends on $y_{i-2}, y_{i-1}, y_i$. Since this is a bounded window into the past and $y_i'$ does not depend on $\hat{x}_{j<i}$, we remain free from temporal error propagation.

|  | Context | LRP | bpp ↓ | PSNR ↑ |
|---|---|---|---|---|
| No previous frames (image codec) | 0 |  | 0.218 | 36.1 |
| 1 previous frame | 1 |  | 0.0907 (-58%) | 36.1 |
| 2 previous frames | 2 |  | 0.0775 (-64%) | 36.1 |
| 2 previous frames and LRP (VCT (Ours)) | 2 | ✓ | 0.0775 (-64%) | 36.8 (+0.7dB) |

Table 2: Ablating how many previous frames we feed to the transformer ("Context"), and whether we use latent-residual prediction (LRP).

# 4 Experiments

## 4.1 Data sets

We train on one million Internet video clips, where each clip has nine frames. We obtained high-resolution videos which we downscale with a random factor (removing previous compression artifacts), from which we get a central 256 crop. Training batches are made up of randomly selected triplets of adjacent frames. We evaluate on two common benchmark data sets: (1) *MCL-JCV* [36, MIT Licence] made up of thirty 1080p videos captured at either 25 or 30FPS and averaging 137 frames per video, and (2) *UVG* [25, CC-BY-NC Licence] containing twelve 1080p 120FPS videos with either 300 or 600 frames each.

**Synthetic videos** We explore three parameterized synthetic data sets that we build by generating videos from still images from the CLIC2020 test set [34, Unsplash licence], (see Fig. 5). Each data set has a parameter $x$ that we vary, and we create 100 videos for each value of $x$. Each video is 12 frames of $512{\times}512$px. We explore: **Shift**, where we pan from the center of the image towards the lower right, shifting by $x$ pixels in each step. **SharpenOrBlur**, where if $x{\geq}0$, we apply Gaussian blurring with sigma $x \cdot t$ at time step $t$. If $x{<}0$, we create videos that get sharper over time by playing a video blurred with $|x|$ in reverse. **Fade**, where we linearly transition between two unrelated images using alpha blending (as in a scene cut). We release the code to synthesize these videos.

## 4.2 Models

We refer to our video compression transformer as **VCT**. We run the widely used, non-neural, standard codec **HEVC** [31] (*a.k.a.* H.265) using the ffmpeg x265 codec in the *medium* and *veryslow* settings, as well as **H.264** using x264 in the *medium* setting. For a fair comparison to our method, we follow previous work [1, 24, 30] in disabling B-Frames, but do not constrain the codecs in any other way. We run the public **DVC** [23] code, and additionally obtain numbers from the following papers: **SSF** [1], which introduced scale-space-flow, an architectural component to support warping and blurring, commonly used in follow-up work, **ELF-VC** [30], to the best of our knowledge the state-of-the-art neural method in terms of PSNR on MCL-JCV, which extends the motion compensation of SSF with more motion priors, **FVC** [17] and **DCVC** [19], both strong models based on warping plus residual coding in a representation space, **RLVC** [41], using ConvLSTMs as a sequence model, and **Liu et al.** [21], who study losslessly transmitting representations using CNNs for temporal entropy modelling. To explore how architectural biases behave on synthetic data, we reproduce SSF, using exactly the same training data as for VCT.

## 4.3 Metrics

We evaluate the common PSNR and MS-SSIM [38] in RGB. We train all models using MSE as a distortion and use $200 \cdot (1 - \text{MS-SSIM}(x, \hat{x}))$ as the training objective in *Stage III* (Tab. 1) to obtain MS-SSIM models.

# 5 Results

## 5.1 Comparison to State of the Art

In Fig. 4, we depict rate distortion graphs for our method and the neural video compression methods introduced in Sec. 4, on MCL-JCV and UVG. Despite the simplicity of our approach, and the fact that we use no motion or warping components, we outperform all methods in both PSNR and MS-SSIM.

## 5.2 Synthetic data

In Fig. 5, we show how the transfomer learns to exploit various types of temporal patterns by applying it to the synthetic data sets introduced in Sec. 4, and reporting the evaluation R-D loss.[5] We compare to HEVC and SSF, which both have explicit support for shifting motion, while SSF also has explicit support for blurring. We expect them to perform well on temporal patterns for which they have corresponding architectural priors. In contrast, VCT has no such priors. For each data set, we explore different values for the parameter $x$ (see Sec. 4), a point in the plot represents the average evaluation loss over the 100 videos created with $x$.

We observe: a) On videos with shifting based motion, VCT obtains $\approx 45\%$ lower R-D loss compared to SSF, which saturates at about $x = 10$, presumably due to the shallow CNN used for flow estimation. Since HEVC supports motion compensating with arbitrary shifts of previous frames, it excels on these kinds of videos. For shifts that are a multiple of 16, the representations shifts by exactly 1 symbol in each step, and VCT matches HEVC. The reason for this is that our encoder is a CNN, so it is only shift-equivariant for shifts which are multiples of the stride (16). Any shift in [1, 15] pixels causes the representation to change in a complex way (cf. [44]). b) For blurring/sharpening, we outperform both HEVC and SSF, despite the latter having explicit support for blurring. Note that the curve for SSF is asymmetric: since it has built-in support for blurring, it gets a $\approx 20\%$ lower RD loss on blurring compared to sharpening. c) VCT learns to handle fading, exhibiting a near-constant RD loss as we increase $x$, in contrast to the baselines, neither of which has a explicit support for fading. SSF is $\approx 20\%$ better than HEVC, possibly due to its blurrying capabilities. For completely static videos $x{=}0$, we observe that VCT is at a slight disadvantage compared to the previous approaches. Overall, we believe that synthetic data can give better insight into the strengths and weaknesses of methods, and hope that future work can compare on these data sets.

## 5.3 Visualizing certainty during decoding

After having seen $k$ tokens in each block, the transformer predicts a PMF $P(t_{k+1}|t_{\leq k}, z_{\text{joint}})$. This induces a joint distribution $P(t_{>k}|\dots)$ over all unseen (not yet decoded) tokens. Intuitively, if the transformer is certain about the future, this distribution will be concentrated on the actual future tokens we will decode. In Fig. 6, we visualize the *sample mean* of this distribution by feeding it through $D$, *i.e.*, we sample $N$ realisations of the unseen tokens in each block, conditioned on the $k$ already decoded ones, for $k \in \{0, 2, 13\}$. In the middle image in Fig. 6, we show what the transformer expects at the current frame, *before decoding any information* ($k = 0$, *i.e.*, 0 bits). We observe that the model is able—to some degree—to learn second order motion implicitly. The next two images shows that as we decode more tokens, the model gets more certain, and the image sharpens.

## 5.4 Ablations

In Tab. 2, we explore the importance of temporal context from previous frames and latent residual prediction (LRP) on MCL-JCV. We start from a baseline that does not use any previous frames, *i.e.*, an image model, used to independently code each frame. Conditioning on one previous frame reduces bitrate by $-58\%$. Using two previous frames yields an additional improvement of $-6\%$. In the final configuration (our model, VCT), which adds LRP, we observe an increase in PSNR of 0.7dB at the same bitrate. We did not observe further gains from more context.

---

[5]$\mathcal{L}{=}r + \lambda d$. To calculate $\mathcal{L}$ for HEVC, we find the quality factor $q$ matching our $\lambda$ via $q{=}\arg\min_q r(q) + \lambda d(q)$, which yields $q{=}25$ for $\lambda{=}0.01$.

|      |       | $T_{\text{sep}}$ and $T_{\text{joint}}$ | $T_{\text{cur}}$ | EC | $D$ | FPS estimate |
|------|-------|------------------------------------------|------------------|---------|---------|---------------|
| Ours | 1080p | 168 ms | 326 ms | 30.5 ms | 168 ms | ≈1.4 FPS |
|      | 720p  | 37.6 ms | 44.8 ms | 17.0 ms | 49.5 ms | ≈6.7 FPS |
|      | 480p  | 18.1 ms | 23.1 ms | 9.02 ms | 23.3 ms | ≈13.6 FPS |
|      | 360p  | 7.3 ms | 14.9 ms | 4.24 ms | 10.1 ms | ≈27.3 FPS |

Table 3: Runtimes of our components. For ours, we use a Google Cloud TPU v4 to run transformers and $D$. Entropy Coding (EC) is run on CPU.

## 5.5 Runtime

To obtain runtimes of the transformers ($T_{\text{sep}}, T_{\text{joint}}, T_{\text{cur}}$) and the decoder ($D$), we employ a Google Cloud TPU v4 (single core) using Flax [16], which has an efficient implementation for autoregressive transformers. We use Tensorflow Compression to measure time spent entropy coding (EC), on an Intel Skylake CPU core. In Tab. 3, we report numbers for $1280{\times}720$px, $852{\times}480$px, and $480{\times}360$px. Since this benchmark is not fully end-to-end, we only report an FPS estimate by calculating $1000/(\text{sum of individual runtimes in ms})$. Note that running $T_{\text{cur}}$ at 720p once only takes ≈2.8ms, but we run it $w_{\text{c}}^2{=}16$ times to decode a frame. To run $T_{\text{joint}}$, we only have to run $T_{\text{sep}}$ once per representation, since we can re-use the output of running $T_{\text{sep}}$ on the previous representation.

Many neural compression methods do not detail inference time and do not have code available, but we copy the results from DCVC [19], FVC [17], and ELF-VC [30], in Table 4.

## 6 Conclusion and Future Work

We presented an elegantly simple transformer-based approach to neural video compression, outperforming previous methods without relying on architectural priors such as explicit motion prediction or warping. Notably, our results are achieved by conditioning the transformer only on a 2-frame window into the past. For some types of videos, it would be interesting to scale this up, or to introduce a notion of more long-term memory, possibly leveraging arbitrary reference frames.

As mentioned towards the end of Sec. 3.2, various different ways to factorize the distributions could be explored, including vector quantization, channel-autoregression, or changing the independence assumptions around how we split representations into blocks.

**Societal Impact**  We hope our method can serve as the foundation for a new generation of video codecs. This could have a net-positive impact on society by reducing the bandwidth needed for video conferencing and video streaming and to better utilize storage space, therefore increasing the capacity of knowledge preservation.

**Acknowledgements**  We thank Basil Mustafa, Ashok Popat, Huiwen Chang, Phil Chou, Johannes Ballé, and Nick Johnston for the insightful discussions and feedback.

|             | Resolution | FPS estimate |
|-------------|------------|--------------|
| Ours        | 1080p      | ≈1.4 FPS |
|             | 720p       | ≈6.7 FPS |
|             | 480p       | ≈13.6 FPS |
|             | 360p       | ≈27.3 FPS |
| DCVC [19]   | 1080p      | ≈1.1 FPS |
| FVC [17]    | 1080p      | ≈1.8 FPS |
| ELF-VC [30] | 1080p      | ≈18 FPS |
|             | 720p       | ≈35 FPS |

Table 4: Comparing decoding speed to other methods. We directly copy reported results from the respective papers, so platforms are not comparable.

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
