# VCT: A Video Compression Transformer – Supplementary Material

## A  Appendix

### A.1  Main Text Image auto-encoder ($E, D$) details

For the results of the main text we use the following architectures for $E, D$. Let C be a $5{\times}5$ convolution with $d_{ED} = 192$ filters and stride 2, followed by a leaky relu activation (with $\alpha = 0.2$). Our encoder $E$ is CCCC. Let T be a $5{\times}5$ transposed convolution with $d_{ED}$ filters and stride 2, also followed by a leaky relu, and let R be a residual block (*i.e.*, R is CC with a skip connection around it). $D$ is RRRRTRRTRRTT, *i.e.*, as we increase in resolution we use fewer residual blocks.

We use the shorthand $4220$ for this, counting the residual blocks between each transpose convolution T. In Fig. 7, we explore $0000$ (no residual blocks) and $2222$. The latter has the same number of residual blocks as our defaults, but uses them in a later stage, making them more expensive (high resolution features).

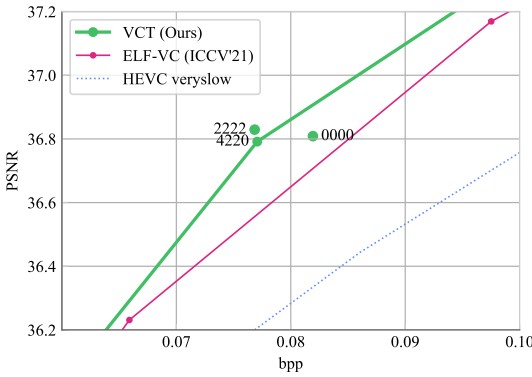

Figure 7: Exploring variants for the decoder $D$, on MCL-JCV.

### A.2  Public Code Release: Simplified Training from Scratch

The official code release at `https://goo.gle/vct-paper` contains a simplified training setup. We only train Stage III (Table. 1), directly **from scratch**, using a LR of $1\text{E}^{-4}$ for 750k steps. We find that the main $E, D$ (see Section A.1) are leading to unstable training when trained from scratch, so we instead use a light-weight architecture from ELIC [15]. The resulting model actually outperforms the architectures presented in the main text, see Fig. 8. We note that we do not need to train a Hyperprior in this setup.

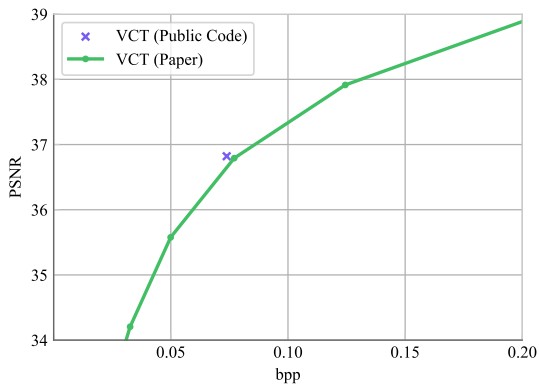

Figure 8: Performance when training from scratch with the public code, on MCL-JCV.

### A.3 Training Set Size

In Fig. 9, we show the effect of dataset size on the loss on MCL-JCV for Ours (VCT) and the CNN based SSF [1]. We observe that VCT benefits from more training data, as has been observed when using transformers in other vision tasks [10]. Note that 50k clips leads to VCT outperforming SSF.

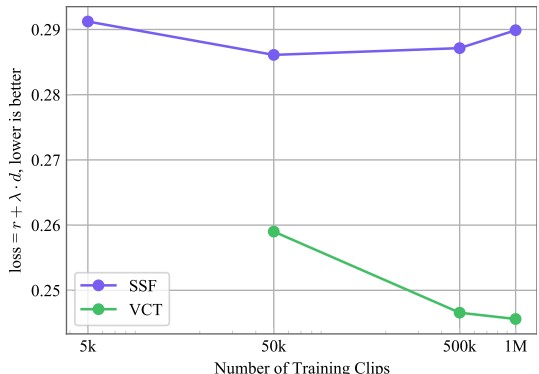

Figure 9: Effect of dataset size on evaluation loss on MCL-JCV. In contrast to the transformer-based architecture, SSF doesn't benefit from a significant increase in the amount of training data.

### A.4 Transformers vs. CNNs

In Fig. 10, we compare VCT against the CNN based method by Liu et al. [21], which studies a similar setting as VCT, but uses CNNs for the temporal entropy model. We provide preliminary results of reproducing Liu et al.'s work using CNNs, trained on our data (purple dot, denoted "Preliminary CNN baseline"). We can see that the baseline obtains a similar rate-distortion performance as the work by Liu et al. Thus, similar to SFF (see Sec. A.3 above), we see that the CNN based approaches do not benefit from additional training data.

The main remaining differences to [21] are: i) they use 1 frame of context (vs. VCT's 2), ii) they rely on CNNs instead of our transformer. We thus plot the model from the ablation study in Table 2, where we only use 1 frame as context, showing that this makes bitrate worse (green cross in Fig. 10, $+18\%$ bitrate increase). From this, we can conclude that the transformer is responsible for the bulk of the remaining gap, *i.e.*, the bitrate increases around 50% when going to a CNN.

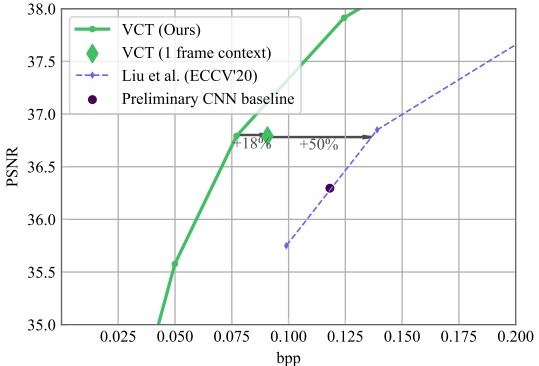

Figure 10: Comparing to Liu et al. [21]

### A.5 Full-sized rate-distortion plots

In the following, we show a larger version of Fig. 4 to aid readability.

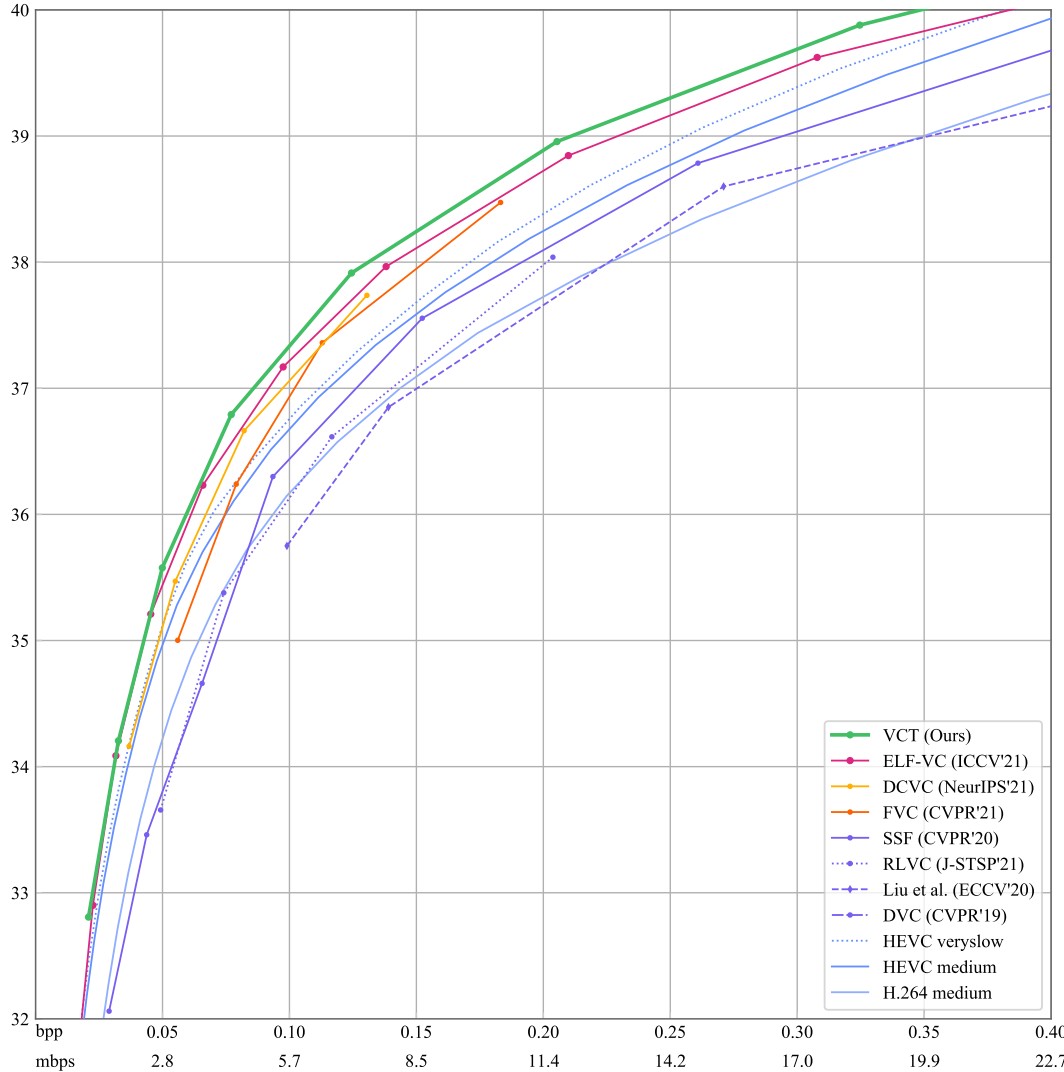

Figure 11: PSNR on MCL-JCV

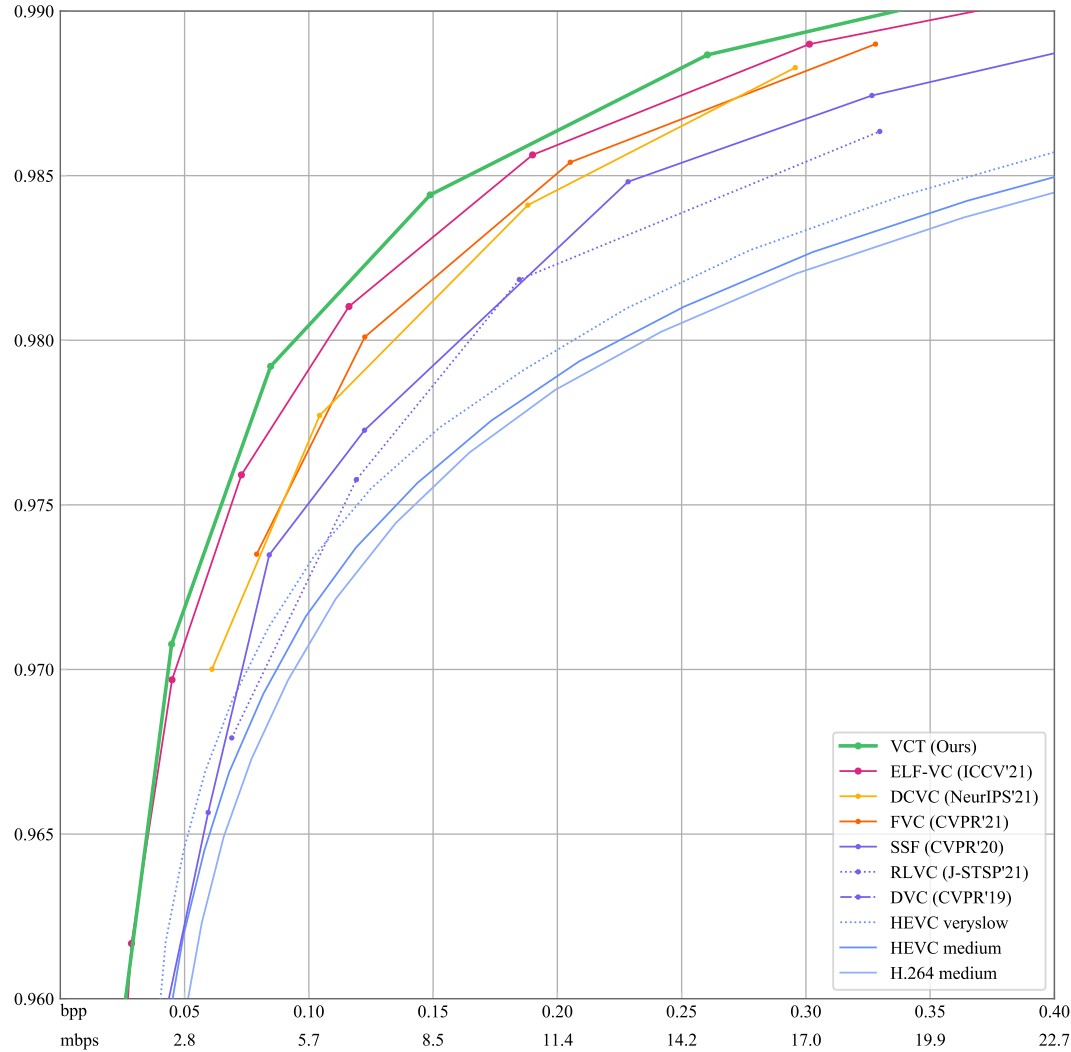

Figure 12: MS-SSIM on MCL-JCV

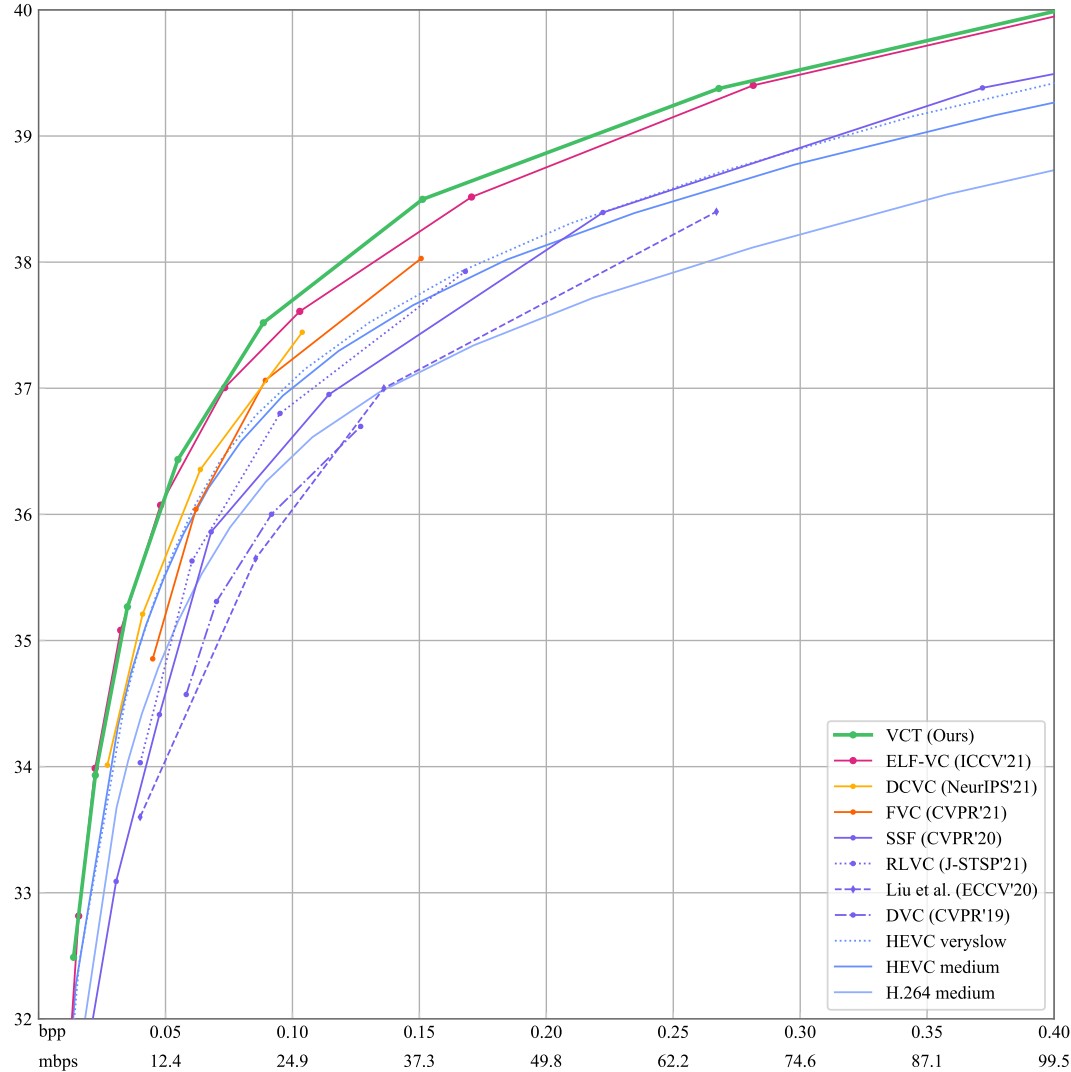

Figure 13: PSNR on UVG

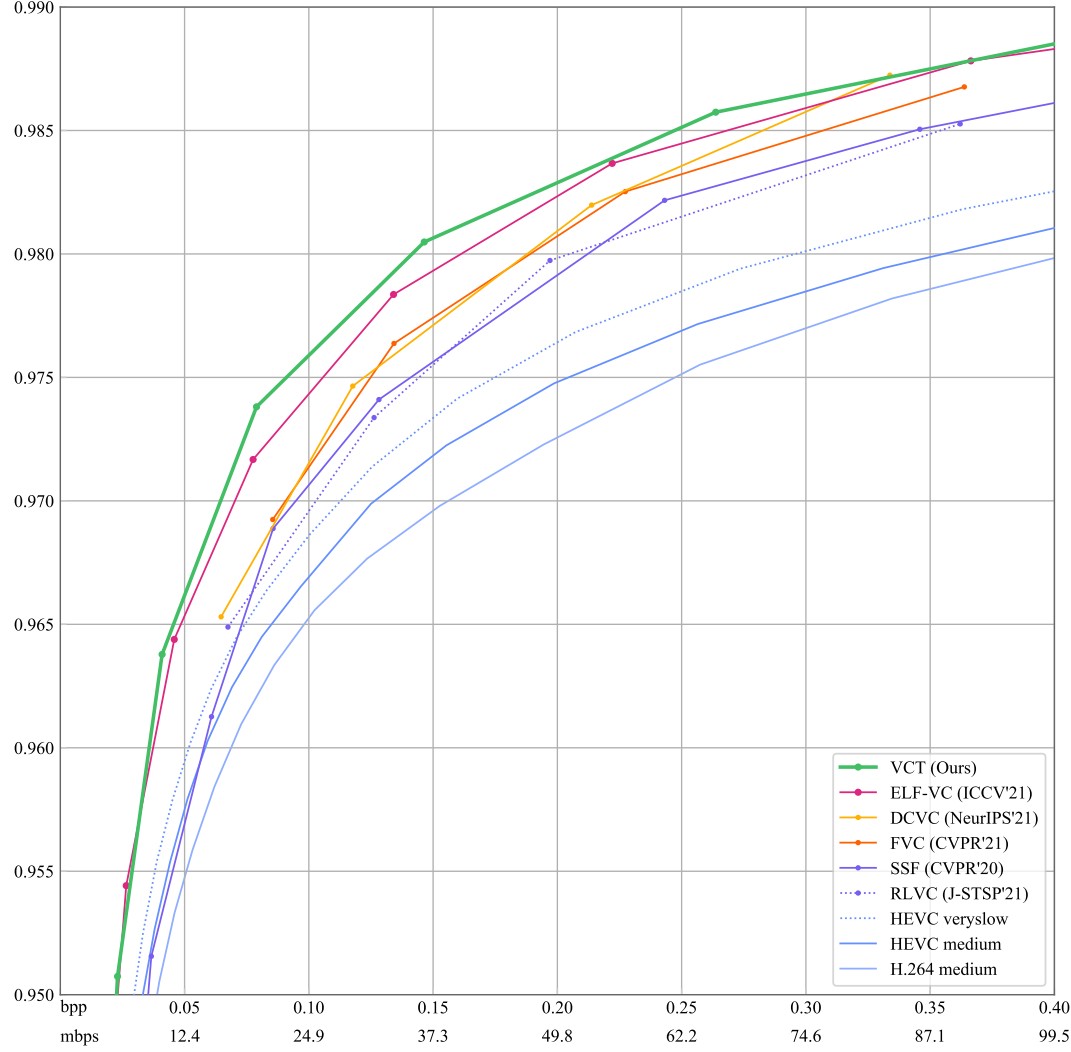

Figure 14: MS-SSIM on UVG