# OpenReview forum: "VCT: A Video Compression Transformer"
_NeurIPS.cc/2022/Conference — NeurIPS 2022 Accept_

### Official Review · Reviewer_xvyj · 2022-07-08

**Rating:** 3
**Confidence:** 5
**Soundness:** 2 fair
**Presentation:** 2 fair
**Contribution:** 1 poor

**Summary:**

This paper uses transformer to capture the relevance between the latent representations of different frames. By using the latent representations of two previous frames as the entropy model input, a good probability estimation can be obtained for that of the current frame. In addition, an improved auto-regressive model is used, where 16 times are needed to decode a frame.

**Questions:**

- The authors use TPU to measure the training and testing time. However, TPU is not available for most researchers. Profiling in NVidia GPU is recommended, and it is easy for the comparison with other related works.
- This paper manually uses the latent representations of 2 previous frames, why not 3 or 4 frames?
-  Each block is divided into 16 parts, why not 4, 9, 25,or 36 parts?


**Limitations:**

No potential negative societal impact

**Strengths And Weaknesses:**


Strength

- This paper investigates the conditional entropy coding-based framework rather than following the mainstream residual-coding based framework where the motion estimation and motion compensation are needed. This is an encouraging attempt.
- Investigate different kinds of synthetic videos including shifting, sharpen/blur, and fading. Some analyses are conducted.

Weakness

- The most issue of this paper is that the novelty is limited. The core contributions have been investigated in other papers.
  - Using the latent representations of previous frames to estimate the probability for that of the current frame has been proposed in [1]. The difference is that one previous frame is used in [1] and two previous frames are used in this paper.
  - Using transformer as the basic unit has been investigated in many neural image codecs, like [2,3,4,5]. Directly applying transformer into neural video codec is straightforward and the novelty is limited.
  - The block-based auto-regressive entropy model is a trade-off between the normal auto-regressive model and checkboard prior model [6]. Actually, the checkboard prior can be regarded as a kind of block-based auto-regressive entropy model, where the block contains two parts. In addition, [7] has investigated a similar idea where each block contains 4 parts. The difference of this paper is that each block contains 16 parts.
 - This paper claims that the motion prediction and warping operations are complex and hand-crafted in mainstream residual-coding based framework, and this paper does not need these designs. The block-based auto-regressive entropy model, LRP, and  3-stage training are also complex and hand-crafted designs.
 - The compression ratio improvement over previous SOTA ELF-VC is limited. The RD-curves are very close. The BD-rate numbers are best presented.
 - Actually, HEVC and H.264 only specify the decoding part and exclude the encoder part. They do not have “veryslow” or “medium” settings. Please use x264 and x265 in the experiment description. In addition, outperforming x264 and x265 is very easy in 2022. Comparisons with JM, HM, VTM are recommended because they represent the best encoders of H.264, HEVC, VVC. The work [8] in 2021 has already compared with VTM.

[1]. Conditional Entropy Coding for Efficient Video Compression, ECCV 2020

[2]. The Devil Is in the Details: Window-based Attention for Image Compression, CVPR 2022

[3]. Entroformer: a transformer-based entropy model for learned image compression, ICLR 2022

[4]. Transformer-based transform coding, ICLR 2022

[5]. Joint Global and Local Hierarchical Priors for Learned Image Compression, CVPR 2022

[6] Checkerboard Context Model for Efficient Learned Image Compression, CVPR 2021

[7] High-Efficiency Lossy Image Coding Through Adaptive Neighborhood Information Aggregation, arXiv 2022.

[8] Versatile Learned Video Compression, arXiv 2021

---

> ### Author Response · Authors · 2022-07-30
> **Response Part 1**
>
> [Response Part 1]
>
> We thank the reviewer on the detailed assessment of our work. We would like to first reiterate that the main focus of this work is to vastly simplify existing learned neural compression algorithms for video and show that our resulting approach outperforms previous state-of-the-art, without having to rely on motion compensation and residual compression.
>
> **Lack of novelty**: We believe that this claim is unsubstantiated and misses the main point of this work mentioned above.
>
> “Using the latent representations of previous frames to estimate the probability for that of the current frame has been explored in [1].”: As we note on L78, the authors of [1] “also losslessly encoded frame-level representations, but rely on CNNs for temporal modeling.” We compare to [1] in Fig. 4 (the approach is referred to as “Liu et al.”), and we can see that their approach performs significantly worse (>1dB in PSNR) demonstrating that our usage of transformers for  temporal modeling is critical for SOTA performance. We emphasize that we show the first neural compression paper that achieves competitive performance in this setting. Furthermore, in contrast to CNNs, Transformers naturally scale to larger temporal contexts by operating on sequences of arbitrary length.
>
> In terms of related work, [2, 3, 4, 5] focus on _image_ compression, which is orthogonal to what we are trying to achieve in this work. These works show that single-frame entropy coding based on the transformer architecture results in competitive models. In fact, given the modular design behind VCT, we could replace the frame encoding with any of these models and potentially improve the results presented in this work. As we stress in the paper (L91-98) that going from images to video is incredibly challenging. The results of conceptually simple approaches which look like straightforward extensions are presented in Table 2 which can be 20x worse wrt coding efficiency. We would also like to note that we cite [1, 3, 4], while [2, 5] appeared after the NeurIPS deadline (CVPR 2022).
>
> Finally, we note that transformer architecture itself was already ported, with minor differences with the original Vaswani et al. paper, to most problems of computer vision. In that sense, all work which applies transformers to new tasks coupled with entropy coding is by design “limited” in novelty. However, we believe that the key behind the success and popularity of models such as the [Vision Transformer](https://arxiv.org/abs/2010.11929) was to actually show that it works in this new domain and can get rid of many of the architectural priors employed previously -- like we saw in our paper. This involves a careful design of the model and the training recipe. It’s not as simple as “just using a transformer”.

---

> > ### Comment · Reviewer_xvyj · 2022-08-08
> > **Comments**
> >
> > Thank the authors for the response. However, my concerns still exist:
> > -  Comprehensive comparisons between the proposed method and Liu et al. “Conditional Entropy Coding for Efficient Video Compression”  ECCV 2020 should be conducted. The overall performance comparison with Liu et al is not enough as the model from Liu et al was trained on 2020. More fair ablation studies should be conducted. Just mentioned by reviewer w64c, is transformer-vs-cnn, training data, number of reference frames, or some other modules?  From the framework comparison, Liu et al. used CNN to estimate the probability from the previous frame and this paper uses a transformer. The additional difference is the block-based auto-regressive entropy model, and the authors never claim it is a core contribution or novelty.
> > -  It is surprising to know that this paper uses 10x more data with respect to SSF. Since the authors have conducted this ablation study, why not show the results?
> > -  The authors do not give a convincing explanation on why not compare with HM or VTM. “Versatile Learned Video Compression” in 2021 has compared with VTM. In 2022, I think this comparison is not too much to ask.

---

> > > ### Author Response · Authors · 2022-08-08
> > > **Response**
> > >
> > > **The overall performance comparison with Liu et al is not enough as the model from Liu et al was trained on 2020.** We compare the rate-distortion tradeoff and significantly outperform their model. Yes, that paper is from 2020, but we fail to see what else is there to do? It seems like the reviewer is implying that we should train all of the models published in the past using the machinery from 2022?
> > >
> > > **transformer-vs-CNN** As we wrote in the reply to reviewer w64c, we are happy to add this comparison.
> > >
> > > **Data** We conducted the study as part of the rebuttal and will add it to the manuscript.
> > >
> > > **VTM** As we write in the rebuttal, "the scope of this work is to show that neural codecs can be significantly simplified and that motion priors are not necessary. "
> > >
> > > We thank the reviewer for detailed comments, but fail to see how the above points are grounds for dismissal of this work. The goal of our research is to sufficiently advance the state of the art, which we believe was clearly demonstrated.

---

> > > > ### Comment · Reviewer_xvyj · 2022-08-08
> > > > **Comments**
> > > >
> > > > - The proposed method in this paper has a very similar idea to Liu et al. Both of them use the latents of previous frames to estimate the probability of the latents of the current frame. Liu et al used CNN and this paper uses transformer. I do not imply that we should retrain all past models in 2022. **But for so similar paper, should we not make comprehensive and fair ablation studies?** Moreover, you use 10x more training data than SSF, is this fair?
> > > > - What is the purpose of developing neural codec?  **Isn't it for higher compression ratio and faster encoding/decoding speed?** Other paper can compare with SOTA traditional codec, why can not your paper compare with it?

---

> > > > > ### Comment · Reviewer_xvyj · 2022-08-08
> > > > > **Reply**
> > > > >
> > > > > The author says that "the scope of this work is to show that neural codecs can be significantly simplified and that motion priors are not necessary. " However, Liu et al. have demonstrated this.

---

> > > > > > ### Author Response · Authors · 2022-08-09
> > > > > > **Response**
> > > > > >
> > > > > > (See response above)

---

> > > > > ### Author Response · Authors · 2022-08-08
> > > > > **Reply**
> > > > >
> > > > > > But for so similar paper, should we not make comprehensive and fair ablation studies?
> > > > >
> > > > > We see the following main differences between our method and Liu et al's method, that should be part of a fair ablation study:
> > > > >
> > > > > 1) Amount of training data
> > > > > 2) 1 vs 2 frames of context
> > > > > 3) CNN vs Transformer
> > > > >
> > > > > To figure out which of these contributes to the significant difference in rate-distortion we see between VCT and Liu et al in Fig. 4, we added an ablation study to the supplementary materials, see the newly added A. 4.
> > > > > We copy the relevant text passage here:
> > > > >
> > > > > ```
> > > > > In Fig. 10, we compare VCT against the CNN based method by Liu et al. [20], which studies a similar setting as VCT, but uses CNNs for the temporal entropy model.
> > > > > We provide preliminary results of reproducing Liu et al.'s work using CNNs, trained on our data (purple dot, denoted ``Preliminary CNN baseline'').
> > > > > We can see that the baseline obtains a similar rate-distortion performance as the work by Liu et al.
> > > > > Thus, similar to SFF (see Sec. A.3 above), we see that the CNN based approaches do not benefit from additional training data.
> > > > >
> > > > > The main remaining differences to [20] are: i) they use 1 frame of context (vs. VCT's 2), ii) they rely on CNNs instead of our transformer.
> > > > > We thus plot the model from the ablation study in Table 2, where we only use 1 frame as context, showing that this makes bitrate worse (green cross in Fig. 10, $+18%$ bitrate increase).
> > > > > From this, we can conclude that the transformer is responsible for the bulk of the remaining gap, ie, the bitrate increases around 50% when going to a CNN.
> > > > > ```
> > > > >
> > > > > We think this improves our paper, and thank you for your suggestion. **If there is a specific additional ablation that would further clarify our method, please let us know.**
> > > > >
> > > > > > Moreover, you use 10x more training data than SSF, is this fair?
> > > > >
> > > > > We note that SSF in the paper uses the same training data as VCT, apologies if this was unclear.
> > > > > Nevertheless, as we show in the added A. 3, this does not have an effect on SSF
> > > > >
> > > > > >  Isn't it for higher compression ratio and faster encoding/decoding speed?  Other paper can compare with SOTA traditional codec, why can not your paper compare with it?
> > > > >
> > > > > Indeed, and we achieve higher compression ratios compared to previous neural codecs, while using a simplified setting. We are open about our runtime in the paper (Table 2), and we believe that our approach advances understanding of neural compression methods to a degree that is sufficient for a NeurIPS paper. If we had a method that is better and faster than VTM, we would try to sell it, not publish it as open research :)
> > > > >
> > > > > > The author says that "the scope of this work is to show that neural codecs can be significantly simplified and that motion priors are not necessary. " However, Liu et al. have demonstrated this.
> > > > >
> > > > > We emphasize that we show the first neural compression paper that achieves competitive performance in this setting, see initial reply.

---

> ### Author Response · Authors · 2022-07-30
> **Response Part 2**
>
> [Response Part 2]
>
> **Comparison to the checkerboard model** [6]: Our usage of block-autoregression contains important novelty: Firstly, previous work [1-7] always relies on additional side information z to condition the entropy model, whereas it is not required in our setting (noted on L156). Secondly, the key idea in [6] is to decompose latents into two sets of squares and code one set given the other. Our approach is only superficially related in that we also use squares, but instead of a two stage global factorization, we _independently_ encode each square, without relying on any neighbors. We can do this thanks to the temporal redundancy of video, i.e., we do not need to rely on global context within the current latent. Similar to [6], [7] (available only on arXiv), also splits latents into sets and codes them in sequence, which again is a different approach from what we do.
>
> Despite all this, we note that we never claim our block-autoregressive scheme is a core contribution or novelty.
>
> **“Motion prediction and warping operations are complex and hand-crafted in mainstream residual-coding based framework, and this paper does not need these designs”**: Note that apart from claiming motion prediction is complex, we also write that previous approaches “constrain themselves to work well only on data that matches the architectural biases” (L18). As a big benefit of removing motion prediction, our model works well on other data, and we see this empirically in our synthetic data (Fig. 5).
>
> In our setup, using autoregressive constitutes a very general factorization of a joint distribution that can hardly be called hand-crafted. It is used in various ML applications such as [language modelling](https://arxiv.org/pdf/1910.10683.pdf) and [image synthesis](https://compvis.github.io/taming-transformers/). Adding LRP corresponds to a single dense layer that maps the transformer features to less channels followed by addition (L204), certainly less complex than the implementation of motion compensation. Finally, our model can be trained end-to-end (i.e., without the stages), but thanks to our modular design it’s easier to just pretrain some components, as done in most recent neural video approaches. This also for fast iteration and research, and is not an important property of the model.
>
> **“The compression ratio improvement over previous SOTA ELF-VC is limited”**: Our empirical results show that our algorithm is competitive and often outperforms SOTA with a simplified design without modeling motion. Thank you for the suggestion to include BD-rate numbers as well, we will compute them and add to the manuscript.
>
> **“Please use x264 and x265 in the experiment description”**: Good point, we will update the manuscript.
>
> **“In addition, outperforming x264 and x265 is very easy in 2022. Comparisons with JM, HM, VTM are recommended [...]”**: The scope of this work is to show that neural codecs can be significantly simplified and that motion priors are not necessary. Our next step is to scale the proposed approach and make it competitive to standard non-neural codecs. We include the ffmpeg implementations of HEVC/H.264 as a well-known baseline.
>
> Questions:
> - Right now, we consider transformers for video compression an early research direction. We chose to report TPU numbers to show that eventually, a VCT-like approach will be feasible for deployment. GPUs are known to lag behind TPUs for attention-based architectures.
> - As mentioned on L280, we do not observe significant gains from using more context. We will add this to the ablation Table 2 to make this more visible.
> - Indeed, other splits would be possible, but we did not explore this for this paper. 16 is natural given 4x4 blocks.

---

### Official Review · Reviewer_yzCS · 2022-07-11

**Rating:** 7
**Confidence:** 4
**Soundness:** 3 good
**Presentation:** 3 good
**Contribution:** 3 good

**Summary:**

The paper proposes a transformer architecture for video compression and demonstrates its favorable performance over prior methods. The method is explicit into two stages: firstly lossily encode each frame independently into a quantized latent space, and secondly use a transformer model to find a probability mass function under which the quantized latent sequence can be efficiently transmitted. The proposed model relies on data-driven prior instead of architectural biases as used in many prior methods.

**Questions:**

1. In table 2, the PSNR of model variants with 0, 1 and 2 previous frames are the same. Does the author have an intuition why this is the case?
2. The paper suggests that more than 2 context frames do not provide further gains in line 280. Is this a result of the type of motion presented in the testing datasets? With larger temporal motions in the testing data, will more frames provide additional benefits, or the it's the capacity of the current model architecture that limits more context information from being efficiently used?
2. A potential extension of the work is to incorporate temporal-aware quantization in the first-stage training instead of processing each frame individually.

**Limitations:**

The limitations and societal impact are addressed.

**Strengths And Weaknesses:**

Strengths:
1. The paper is clearly written and easy to follow.
2. The method achieves favorable empirical performance on standard benchmarks.
2. The paper discusses the the type of temporal patterns that the model learns to exploit by examining synthetic videos. The discussion provides a good understanding of the model's capability when applied to data sets with different motion patterns.

Weaknesses:
1. The comparisons of model size and inference time with baseline methods are not listed.
2. The paper provides an intuitive explanation in line 108-110 of why the second stage is advantageous compared to the naive solution. The hypothesis could be tested on synthetic datasets, which will further strengthens this claim and verifies if this is indeed what the type of inductive bias that the model learns from the data.

---

> ### Author Response · Authors · 2022-07-30
> **Response**
>
> We thank the reviewer for recognizing and acknowledging the strengths of the proposed approach. We will address the raised questions and concerns here.
>
> 1. **Parameter Counts/Runtimes**: Here is a preview of the requested parameter counts, we will add the full information to the paper: VCT: 114M // ELF-VC: 10.7M // FVC: 26M. Many neural compression methods do not detail inference time and do not have code available, but we will add the following from the top methods in Fig. 4: ELF-VC with speed as a main focus reports an impressive 35 FPS decode time on 720p, VCT obtains 6.7 FPS on 720p, DCVC reports 857ms inference on 1080p (i.e. 1.1FPS, although it's unclear if encoding is included), FVC reports 548ms on 1080p (1.8 FPS). We hope that future work will follow VCT in more clearly reporting speed and, more importantly, releasing code.
> 2. **Naive solutions**: We note that Table 2 contains one naive solution, i.e., not considering previous frames, while the other naive solution, namely using a uniform distribution, is discussed in the main text. Given that the latter always requires $\log_2 |\mathcal S|$ bits per symbol (9MB per HD frame), or 4.5bpp, it is 20x worse than the ablation presented in Table 2. We are unsure whether this covers your question or you have another naive solution in mind?
>
> Regarding your questions:
> - **PSNR of ablations**: Without LRP, the PSNR at frame $i$ only depends on $y_i$, which we losslessly transmit. Hence, using fewer frames only increases the bitrate but doesn’t affect PSNR. As a result, even if the model is underfitting, and e.g. predicing $p \sim U$, we would still achieve 36.1 dB PSNR, only the coding efficiency would decrease significantly.
> - **More context does not help**: (a) Your point regarding capacity is well aligned with what we observe in practice with large transformer models -- the learning problem becomes more challenging with the increase in the input sequence length required to model more contextual information. (b) At a short-temporal scale, the previous frame encapsulates most of the information necessary to predict the current frame. The main exception is the presence of occlusions, but on average one needs to see many more frames to handle occlusions (i.e. an object might be occluded for 30+ frames and then become visible again).
> - **Temporal-aware quantization**: This is a very good point and something we also considered during model development and will consider in future work. In principle, it seems sub-optimal to use independent encoders, but it does simplify the training setup.

---

> > ### Comment · Reviewer_yzCS · 2022-08-08
> > **Thank you for your responses**
> >
> > Thank you for the clarifications which addressed my concerns and I believe my current rating is consistent with the updated version. The proposed method has a reasonable inference speed. It uses a larger model compared to baselines and I believe this should be discussed in the limitations, but I don't think it under-weighs the contribution of this paper. The paper presents a cleanly designed model for video compression with favorable empirical performance and clearly presented ablation studies. It is a good fit for the venue in my opinion.

---

### Official Review · Reviewer_w64c · 2022-07-11

**Rating:** 7
**Confidence:** 4
**Soundness:** 4 excellent
**Presentation:** 3 good
**Contribution:** 3 good

**Summary:**

The authors proposed a video compression transformer (VCT) for neural video compression. Input frames are mapped to representations, and a transformer is used to model the dependencies. The VCT model achieves SOTA performance on MCL-JCV and UVG datasets. Both theoretical and empirical results are promising and will benefit future research.

**Questions:**

1. What’s the HEVC setting in Figure 5? Since SSF is not using a pretrained optical flow model, I wonder what other models (e.g., DVCPro or DCVC that builds on SPyNet) would behave on the shift sequences. It seems that learning-based models achieve superior benchmark performance than HEVC by handling distortions such as sharpen/blur/fade better, but not as well to shifts. It’s unclear what’s the cost of not using motion compensation in the VCT framework, whether the cost is negligible or if the transformer predicts good enough distributions such that an ME module is unnecessary.

2. It seems that the VCT handles shifts better when the shift is the same size as the patch size. Are the authors aware of/considering any approach to improve this problem?

3. VCT and SSF are trained on different data from most other works. Although larger-scale training has been proved necessary on various language/vision tasks, is there an ablation study on the size of the training data for VCT? This would help to understand the limitation of the model, its scalability, and its potential for future work.

4. A less important question: have the authors considered the HEVC dataset and other video compression metrics (e.g., AVQT or perceptual-based metrics)? As VCT is a novel architecture, it would help to understand how it compares with previous works as well as decide future directions.

**Limitations:**

In this work the authors proposed VCT for neural video compression. The model design is novel and elegant and the model architecture follows most common designs. It helps to show the efficacy of the proposed framework and many improvements are also possible (as summarized in Section 6). Some possible limitations in terms of motion compensation and training data are mentioned in Question 1 and 3.

**Strengths And Weaknesses:**

**Strengths**:

1. The proposed framework is simple and effective. The frames are decoded from the quantized representation, which prevents the problem of error propagation for residual coding-based video compression methods.

2. The initial frames are decoded with zero padding for y_i-1 and y_i-2 so the I-frame and P-frame models in residual coding-based framework are unified into one model. This design large simplifies the video compression framework and will benefit future research.

3. In order to speed up the execution, the authors assumed independence between blocks and only overlapping tokens from i-1 and i-2 are used. This seems a bold assumption at the beginning but proved effective.

**Weaknesses**:

1. Most modules follow the most common designs and are not curated for this specific model. Some weaknesses/improvements are considered by the authors in Section 3.2-Independence and Section 6.

2. Other weaknesses and concerns are detailed in the ‘Question’ section.

---

> ### Author Response · Authors · 2022-07-30
> **Response**
>
> We thank the reviewer for the diligent review and address the remaining questions here.
>
> - **“Most modules follow the most common designs and are not curated for this specific model”**: Indeed, this is by design -- we believe that neural compression models should build on generic components which are being independently improved by other researchers (e.g. in computer vision), which we know how to train at scale, and how to debug. We believe that this is one of the main advantages of our model.
> - **“What’s the HEVC setting in Figure 5?”**: We use the medium setting, disabling B-Frames (see L255-L228).
> - **“I wonder what other models [...] would behave on the shift sequences”**: That’s a good question! We assume that methods based on powerful pre-trained optical flow networks will fall between SSF and HEVC. However, we do not have access to these codebases, but we are hopeful that such comparison will be done after our model and synthetic data loader are released.
> - **“It’s unclear what’s the cost of not using motion compensation in the VCT”**: Fig 4 shows that for _natural_ videos motion compensation is not critical, as we outperform motion-based approaches on MCL-JCV and UVG. In fact, Fig 5 shows for videos which are not explained by translation-based motion, one actually benefits from not being constraint to use motion compensation as a main component. Whether combining VCT with a motion-based component yields additional benefits will be an interesting investigation for future work.
> - **“It seems that the VCT handles shifts better when the shift is the same size as the patch size.”**: This observation is correct and we also comment on it in L255. The reason for this is that our encoder is a CNN, so it is only shift-equivariant for shifts which are multiples of the stride (16). Specifically, if the input shifts by exactly 16 pixels, the representation shifts by exactly one symbol. Any shift in [1, 15] pixels causes the representation to change in a complex way (cf. [Making Convolutional Networks Shift-Invariant Again](https://arxiv.org/abs/1904.11486)).
> - **Ablation study on the size of the training data**: We performed this ablation study and found that VCT requires approximately 10x more data with respect to SSF. We assume that it is possible to obtain similar performance as we report in the paper with less data and augmentation strategies such as sub-sampling temporally, reversing videos, shifting videos, etc., following works that use transformers for other vision tasks. That being said, given the unsupervised task of compression, training data is abundant and we leave this specific exploration for future work.
> - **HEVC dataset and metrics**: Regrettably, we do not have access to the HEVC dataset and metrics. We do believe that proper benchmarking is indeed a cornerstone of science, and to help alleviate this challenge we will be open-sourcing our synthetic data set. Regarding additional metrics, an additional advantage of the modular design of VCT is that the image encoder/decoder pair can be swapped-out. Thus, using other metrics or training a generative decoder are easy to explore.

---

> > ### Comment · Reviewer_w64c · 2022-08-07
> > **Feedback**
> >
> > Thank the authors for the response, including more details and results on VCT. I appreciate the contributions in this work -- both the proposed framework and other experimental results as detailed in the original review. After careful consideration, I decided to adjust my rating based on the following reasons:
> > * **Concerns regarding performance on shift sequences.** Fig 5 shows that VCT underperforms HEVC medium on shift sequences despite superior performance on natural sequences. This points out an important limitation of VCT in handling motions. Fig 5 also lacks a comparison with other SOTA methods using motion estimation networks -- including DVC and DVC Pro which are publicly available and can be easily reproduced.
> >   * The idea of VCT + motion-based components is not a trivial extension and is questionable as simplicity (e.g., not using motion compensation) is a main contribution of VCT. Although benchmark performance on challenging datasets is important, we should also investigate the limitations on sequences with large motion, including synthetic shift sequences.
> > * **Some limitations in training.** Quantitative results in Fig 4 are not necessarily a fair comparison. SSF and VCT use extra private training data. This limits the impact of VCT on future research as it is difficult to reproduce and follow.
> > * **Concerns about the novelty of "using the latent representations of previous frames to estimate the probability for that of the current frame", as pointed out xvyj.** What's the key of VCT to outperform this previous model? Is transformer-vs-cnn, training data, number of reference frames, or some other modules? Some ablation study experiments to break down the performance gap would help to understand the model, beyond benchmark performance.

---

> > > ### Author Response · Authors · 2022-08-08
> > > **Response**
> > >
> > > Thanks for your kind words regarding contributions and for the reply. Regarding your additional concerns:
> > > - **Shift Sequences**: We are in agreement, except maybe that shift sequences are a very specific "unnatural" sequence. We actually interpret Fig 5 in the opposite way: It is impressive how much better we do than SSF despite no explicit motion prediction and warping.
> > > - **Training data**: As mentioned, training data for our unsupervised compression task is abundant. We note that our "private" data is simply downloaded from public video streaming platforms. An example available large dataset today would be [YT8m](https://research.google.com/youtube8m/download.html).
> > > - **CNNs**: From preliminary results we understand that the gap is from "transformer-vs-cnn", since we see in the paper that more reference frames gives limited gains and we know that our CNN based baseline (SFF) did not benefit from more training data. Given points 1 and 2, would adding a transformer vs CNN comparison make you reconsider the rating?

---

> > > > ### Comment · Reviewer_w64c · 2022-08-08
> > > > **Re: A.3**
> > > >
> > > > * What's the "evaluation loss" in A.3/Figure 9 calculated on? Training data, validation data, or UVC/MCL-JCV?
> > > > * What's the model size of SSF and VCT in terms of parameters?
> > > > * Please also fix the scale of x-axis in the future version -- log scale or uniform scale.

---

> > > > > ### Author Response · Authors · 2022-08-08
> > > > > **Response**
> > > > >
> > > > > - It's on MCL-JCV, we updated the supplementary.
> > > > > - VCT: 114M, SSF: 27M parameters
> > > > > - We updated the supplementary to use a log scale, thanks for pointing this out.

---

> > > ### Author Response · Authors · 2022-08-08
> > > **Lack of Novelty: Additional Reply**
> > >
> > > As part of our reply to Reviewer xvyj, we wrote the following, which we highlight again here:
> > >
> > > We see the following main differences between our method and Liu et al's method, that should be part of a fair ablation study:
> > >
> > > 1) Amount of training data
> > > 2) 1 vs 2 frames of context
> > > 3) CNN vs Transformer
> > >
> > > To figure out which of these contributes to the significant difference in rate-distortion we see between VCT and Liu et al in Fig. 4, we added an ablation study to the supplementary materials, see the newly added A. 4.
> > > We copy the relevant text passage here:
> > >
> > > ```
> > > In Fig. 10, we compare VCT against the CNN based method by Liu et al. [20], which studies a similar setting as VCT, but uses CNNs for the temporal entropy model.
> > > We provide preliminary results of reproducing Liu et al.'s work using CNNs, trained on our data (purple dot, denoted ``Preliminary CNN baseline'').
> > > We can see that the baseline obtains a similar rate-distortion performance as the work by Liu et al.
> > > Thus, similar to SFF (see Sec. A.3 above), we see that the CNN based approaches do not benefit from additional training data.
> > >
> > > The main remaining differences to [20] are: i) they use 1 frame of context (vs. VCT's 2), ii) they rely on CNNs instead of our transformer.
> > > We thus plot the model from the ablation study in Table 2, where we only use 1 frame as context, showing that this makes bitrate worse (green cross in Fig. 10, $+18%$ bitrate increase).
> > > From this, we can conclude that the transformer is responsible for the bulk of the remaining gap, ie, the bitrate increases around 50% when going to a CNN.
> > > ```
> > >
> > > ---
> > >
> > > We hope this addresses your concern on "Some ablation study experiments to break down the performance gap would help to understand the model"

---

> > > ### Comment · Reviewer_w64c · 2022-08-09
> > > **Feedback after discussions**
> > >
> > > Thank the authors for providing extra results on VCT. They are an addition to the paper. I decide to keep my current evaluation.
> > >
> > > The ablation study on training data and the comparison with Liu et al. [20] are essential for rigorous experimental results. Based on the current ablation, some more ablation studies would help: (these are time-consuming experiments so they are just suggestions for future revision)
> > > * **Ablation on training data**: consider adding DVC or DVC Pro as another baseline -- what would other optical flow-based methods perform in this figure?
> > > * **Ablation on comparison with Liu et al.**: I conjecture that the gap between "Preliminary CNN baseline" and VCT is still non-trivial. I would recommend some more experiments on this.
> > >
> > > Finally, I do not agree with some of reviewer xvyj's comments. I acknowledge the concerns about the novelty of this work since there were some related ideas in the literature. I don't think we should completely disregard some novel modules proposed in this work. Some of these concerns (e.g., novelty and comparison with other works) don't quite fit into the criteria of a clear reject (3).

---

### Official Review · Reviewer_7MPC · 2022-07-25

**Rating:** 5
**Confidence:** 2
**Soundness:** 3 good
**Presentation:** 2 fair
**Contribution:** 2 fair

**Summary:**

The authors tackle the problem of video compression using transformers. The proposed architecture doesn't include specific biases and priors like motion prediction, blurring, etc. Transformers are used to predict the distribution of the future representation given the earlier video frames. The idea is to exploit the temporal redundancy across frames and the spatial consistency within frames which can help in entropy coding. They show good results using transformers on standard video compression data sets (MCL-JCV, UVG, synthesized videos from CLIC2020) as compared to previous approaches.

**Questions:**

It may help if the authors can tell how the entropy coding is done in the current context (see weaknesses above). Also if the authors can address any other weaknesses mentioned above then it can help in better understanding of the approach.

**Limitations:**

Yes the authors have adequately addressed the limitations and potential negative societal impact of their work.

**Strengths And Weaknesses:**

Strengths:
* There is no explicit biases or priors, which may help the proposed architecture to generalize better across datasets
* The authors introduce independence assumptions among the blocks of each video frame which shrinks the attention matrix and enables parallel execution on subsets of the video frame.

Weaknesses:
* The paper is not easily readable as there are several inconsistencies as pointed below
* In Fig 1, it seems that y_i is fed to the Masked Block-Autoregressive Transformer, but from the later text, it seems it only previous tokens of y_i are fed to get the probability of the current token of y_i.
* line 207: Even though it is a bounded window, if z_cur is used to decide y'_i then y'_i does indirectly depend on all previous encodings which can propagate errors. This counters the claim made in line 39 that the architecture doesn't propagate temporal errors.
* Entropy coding: The distribution of the next token can help identify the number of bits to encode the token. But it is not clear how one of the S symbols can be arbitrarily encoded into that many bits.
* Availability of code or at least pseudo-code in the appendix could have helped understand how entropy coding is done in the current approach
* Table 3: To evaluate better, the runtime of the baselines should also be presented

Minor:
* Figure 4: bits per pixel is not cited
* line 189: what is distortion loss?
* line 280: Even though no further gain is observed from more context, it may help if it is shown in the Table 2.

Typos:
* line 36: '2.1.1' instead of '2.2.1'
* line 56: 'for' is redundant
* line 108: The output of T_cur should be z_cur, but the figure has z_joint.
* Eqn 2: round(y_i) should be round(\tilde(y)_i) ?
* line 192: integration is done over du ?

---

> ### Author Response · Authors · 2022-07-30
> **Response**
>
> We thank the reviewer for careful reading and suggestions. We will update the manuscript to clarify the minor issues and typos, and discuss the major items here:
>
> **Entropy Coding**: We opted for a higher-level description due to space constraints, but we can definitely provide more detail for the benefit of the reader by providing a definition and a pointer to [Huffman coding](https://en.wikipedia.org/wiki/Huffman_coding), followed by a concrete instance with a few symbols, such as:
>
> > Let us assume symbols {A, B, C}, and P(A) = ½, P(B) = ¼, P(C) = ¼. Now, an optimal code is to assign the bitstring 0 to A, 10 to B, and 11 to C. Then, the example sequence AABC would become 001011. Note that A takes exactly $-log_2(P(A)) = 1$ bits, and B, C take 2 bits each. The entropy coding scheme we use works similarly conceptually, but is also optimal if your probabilities are not powers of 2: [Arithmetic Coding](https://en.wikipedia.org/wiki/Arithmetic_coding).
>
> Note that we will release code to reproduce the entire paper, including the entropy coding part. Please let us know whether this is sufficient.
>
>
> **Propagating errors**: The claim on L39 is correct as we transmit representations using the following steps:
> - First, given $N$ frames, we extract all quantized representations $y_1, y_2, \\dots, y_N$ independently using the encoder $E$ (L32). Then, we use the transformer to _losslessly_ transmit these by predicting distributions and using arithmetic coding (L35). We emphasize that if the transformer is bad at predicting distributions, it may not be efficient in terms of the number of bits used to encode $y_i$, but the transmission will nevertheless be_lossless_.
> - As described on L112-119, the receiver can use the transformer to recover $y\_1, y\_2, \\dots, y\_N$. It then applies LRP, calculating $y’\_i = y\_i + z\_\\text{cur}$. Note that $z\_\\text{cur}$ is a function of $y\_i, y\_{i-1}, y\_{i-2}$ and not a function of $y’\_i$. Thus, we never use the previous $y’\_{i-1}, y’\_{i-2}, \\dots$ to calculate $y’\_i$, we only use  $y\_i$, which does not contain errors. You can imagine this as a graph where information only flows in one direction, there is no connection from $y’\_i$ to the $i+1$-th reconstruction.
>
>
> Hence, the i-th reconstruction does not depend on all previous encodings and the errors are bounded, as claimed in the text.
>
> **Table 3 and Fig 1**: Regarding Fig 1, you are absolutely correct, only previous tokens of $y\_i$ are processed to calculate the probability of the current token. However, our aim was to show a high-level overview of the information flow with respect to all tokens. We do agree that striking the right balance between high-level overviews and exact low-level operations is challenging, and could provide an alternative visualization of only a single step (where $y\_{i,<t}$ is at the input, and $P(y\_{i,t}|y\_{i,<t},y\_{i-1},y\_{i-2})$ at the output) if that would improve the clarity. At the moment we will improve the caption to clarify this potential discrepancy. Finally, we will update Table 3 to include additional runtime information.

---

> ### Author Response · Authors · 2022-08-09
> **Updated Manuscript**
>
> We have updated the manuscript with the modified figures and addressed the typos. Let us know if Fig. 1 in particular is an improvement from your point of view.

---

### Author Response · Authors · 2022-08-06
**Need further clarification?**

Thank you for your constructive and detailed comments. We believe that we have addressed the raised concerns below. Are there any remaining points to clarify? We would be happy to engage in discussions if needed.

---

### Author Response · Authors · 2022-08-08
**Updated manuscript**

We have updated the main manuscript with blue text highlighting new lines.

- To `7MPC`: We have updated the manuscript to fix the typos, updated Fig. 1 (please take a look), updated Fig 2, added Section A.2 (in the supplementary material) on entropy coding, and updated Table 3, and added a new Table 4.
- To `w64c`: We have extended the note on shifts in the updated manuscript, and added Section A.3 to the appendix on data ablations, and Section A.4 with a comparison to Liu et al.
- To `yzCS`: We have added a new Table 4 with runtime comparisons
- To `xvyj`: We have added Section A.4 with a comparison to Liu et al.

---

### Meta-Review · Area_Chair_4cuo · 2022-08-24

**Recommendation:** Accept
**Confidence:** Certain

**Metareview:**

This paper uses transformers for video compression, using less components compared to competing methods. Video compression is an important application in machine learning, and the use of transformers is well-timed w.r.t. generally strong interest in the architecture. There were some concerns over clarity of presentation, as well as issues with some of the experimentation, which the reviewers and authors seem to mostly seem to have been able to work out. There is one exception in one reviewer: the authors seem to have worked very hard to address all of this reviewer's concerns, but said reviewer did not adjust their score at all. In addition there was disagreement on some of that reviewers more prominent points.

So I recommend acceptance of this paper.

Overall, w64c stood out as an exceptional reviewer, as they engaged beyond their own review and actively both with the authors and the other reviewers. xvyj brought up some good points, but they were almost tyrannical towards the authors and never gave back in terms of score when the authors clearly satisfied their concerns.

**Award:**

No

---

### Decision · Program_Chairs · 2022-09-14

Accept